# Molecular basis of Mg²⁺ permeation through the human mitochondrial Mrs2 channel

Ming Li [1,4], Yang Li [1,4], Yue Lu [1], Jianhui Li [1], Xuhang Lu [1], Yue Ren [1], Tianlei Wen [1], Yaojie Wang [1], Shenghai Chang [2,3], Xing Zhang [2,3], Xue Yang [1] ✉ & Yuequan Shen [1] ✉

Mitochondrial RNA splicing 2 (Mrs2), a eukaryotic CorA ortholog, enables Mg²⁺ to permeate the inner mitochondrial membrane and plays an important role in mitochondrial metabolic function. However, the mechanism by which Mrs2 permeates Mg²⁺ remains unclear. Here, we report four cryo-electron microscopy (cryo-EM) reconstructions of *Homo sapiens* Mrs2 (hMrs2) under various conditions. All of these hMrs2 structures form symmetrical pentamers with very similar pentamer and protomer conformations. A special structural feature of Cl⁻-bound R-ring, which consists of five Arg332 residues, was found in the hMrs2 structure. Molecular dynamics simulations and mitochondrial Mg²⁺ uptake assays show that the R-ring may function as a charge repulsion barrier, and Cl⁻ may function as a ferry to jointly gate Mg²⁺ permeation in hMrs2. In addition, the membrane potential is likely to be the driving force for Mg²⁺ permeation. Our results provide insights into the channel assembly and Mg²⁺ permeation of hMrs2.

The magnesium ion (Mg²⁺) is an intracellular divalent cation that plays a vital role in many physiological activities in living organisms. It often functions as a cofactor and activator of ATP molecules to facilitate enzymatic reactions[1,2]. Recently, Mg²⁺ has been shown to be a second messenger that regulates CD8⁺ T-cell activity in the immune system[3] and transduces signals through metabolic circuits[4]. Therefore, maintaining Mg²⁺ homeostasis is important to living organisms[5]. Mg²⁺ imbalance can lead to serious diseases, including cardiovascular disease, hypertension, obesity, Parkinson's disease and cancer, etc[2,6,7].

Mg²⁺ transmembrane transport is sustained by Mg²⁺ channels and transporters[5,8]. Eukaryotic mitochondrial RNA splicing 2 (Mrs2), a homolog of prokaryotic CorA, anchors to the inner mitochondrial membrane and mediates the influx of Mg²⁺ into the mitochondrial matrix. It is essential for mitochondrial Mg²⁺ homeostasis[9–11]. In human cells, Mrs2 deficiency leads to marked changes in mitochondrial morphology and severely dysregulates mitochondrial physiology[12,13]. On the other hand, cells overexpressing Mrs2 show increased mitochondrial and whole-cell uptake of magnesium ion[14].

Several prokaryotic CorA structures, such as those in *Thermotoga maritima* (TmCorA)[15–18] and one archaea CorA structure in *Methanocaldococcus jannaschii* (MjCorA)[19], have been reported to date. CorA proteins typically assemble as homo-pentamers with large funnel-shaped intracellular domains (ICDs) and transmembrane domains (TMDs). Based on structural models and functional studies, four main models of CorA gating mechanisms have been proposed: an iris-like opening[20,21], stem-helix rotation[19,22], considerable helical torque[23] and three distinct motions[24]. In 2016, the cryo-EM structure of TmCorA showed that the absence of Mg²⁺ results in the disruption of intracellular subunit symmetry in CorA and expansion of the ion-conducting pore[18]. Subsequently, molecular dynamics (MD) simulations showed that the asymmetric motion of the monomer enables the rotation of helix α7 and the cytoplasmic subdomain to open the gate[25]. Recent studies have shown that CorA is in a dynamic equilibrium between different states of symmetry and asymmetry, and that the conducting state increases upon activation[26].

[1]State Key Laboratory of Medicinal Chemical Biology and Frontiers Science Center for Cell Responses, College of Life Sciences, Nankai University, Tianjin 300350, China. [2]Department of Biophysics and Department of Pathology of Sir Run Run Shaw Hospital, School of Medicine, Zhejiang University, Hangzhou 310058, China. [3]Center of Cryo Electron Microscopy, Zhejiang University, Hangzhou 310058, China. [4]These authors contributed equally: Ming Li, Yang Li. ✉e-mail: yangxue@nankai.edu.cn; yshen@nankai.edu.cn

However, the mechanism of $Mg^{2+}$ transport via the CorA family in eukaryotic systems remains unclear. To better elucidate the mechanism through which $Mg^{2+}$ moves through the eukaryotic CorA homolog Mrs2, we report on several cryo-EM reconstructions of human Mrs2 (hMrs2) under different conditions. Structural analysis, MD simulations and mitochondrial $Mg^{2+}$ uptake assays showed that hMrs2 may function as a membrane potential-driven $Mg^{2+}$ channel enabling permeates $Mg^{2+}$ facilitated by $Cl^-$ ions.

## Results

### Structure determination

hMrs2 proteins were expressed transiently in HEK-293F cells and purified in the presence or absence of $Mg^{2+}$ or at different pH values using the detergent glycol–diosgenin (GDN) and then concentrated for cryo-EM sample preparation. All datasets were collected using a Titan Krios electron microscope operated at an accelerating voltage 300-kV with a Gatan K2 Summit or a Thermo Falcon 4 direct-electron-counting detector. Details about the data collection and data processing are shown in Supplementary Figs. 1–4. The hMrs2 structure was reconstructed from the dataset in the presence of 20 mM $Mg^{2+}$, pH 8.0 (hereinafter referred to as hMrs2-Mg); from the dataset without the addition of external $Mg^{2+}$ or EDTA, pH 8.0 (named hMrs2-rest); from the dataset in the presence of 1 mM EDTA, pH 8.0 (named hMrs2-lowEDTA); or from the dataset in the presence of 5 mM EDTA, pH 6.8 (named hMrs2-highEDTA). The final 3D reconstructions of hMrs2-Mg, hMrs2-rest, hMrs2-lowEDTA and hMrs2-highEDTA were determined at

overall resolutions of 2.6 Å, 2.9 Å, 2.5 Å and 2.7 Å, respectively (Supplementary Table 1). The cryo-EM maps are sufficient to resolve most amino acid side chains (Supplementary Figs. 1–4).

### Overall structure

The overall structure of the hMrs2 channel has a cone shape, and it is assembled into a pentamer with a central pore in the middle (Fig. 1). The hMrs2 height is estimated to be approximately 131 Å, and the width of the intermembrane space side and that of the matrix side are approximately 46 Å and 100 Å, respectively (Fig. 1a, b). The pentameric hMrs2 structure can be further separated into two parts: the N-terminal funnel domain within the matrix and the C-terminal transmembrane domain in the inner mitochondrial membrane. From both the intermembrane space view and matrix view, the approximate fivefold symmetry of the pentameric hMrs2 structure is observed (Fig. 1c, d). The overall structures of hMrs2-rest, hMrs2-lowEDTA and hMrs2-highEDTA were very similar to that of hMrs2-Mg (Fig. 1e–g). We did not observe any significant conformational changes among the four structures.

Compared with previously published CorA family protein structures (*T. maritima* CorA, TmCorA; *Methanocaldococcus jannaschii* CorA, MjCorA; *Escherichia coli* ZntB, EcZntB; and *Pseudomonas aeruginosa* ZntB, PaZntB), the hMrs2 channel pentameric architecture is similar[15–17,19,27,28], albeit that its sequence homology with them is low (Supplementary Fig. 5). The Cα atom root-mean-square deviation (RMSD) values obtained after superposition of hMrs2 with each of the

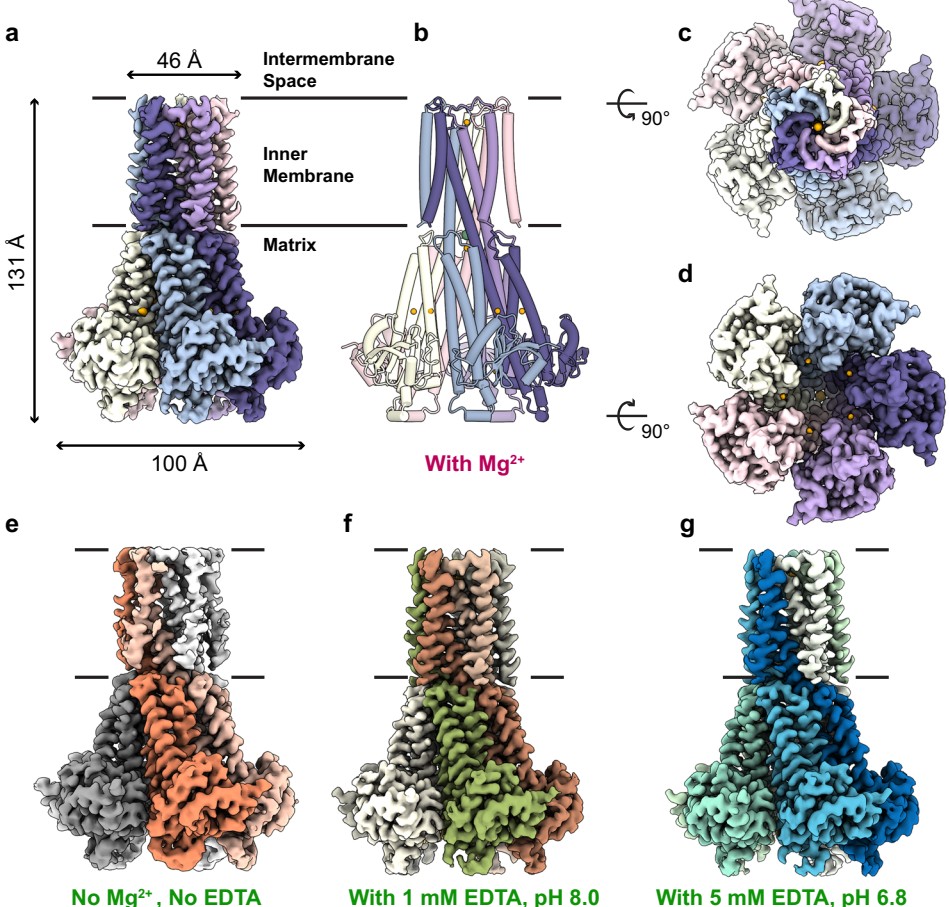

**Fig. 1 | Overall structure of hMrs2 under different conditions.** Side (**a**), top (**c**) and bottom (**d**) views of the cryo-EM reconstruction of the hMrs2-Mg pentamer. $Mg^{2+}$ is shown in orange. **b** Side view of a cartoon representation of the hMrs2-Mg pentamer. **e**–**g** Side view of the cryo-EM reconstruction of the hMrs2 pentamer under three different conditions. **e** hMrs2-rest (no $Mg^{2+}$, no EDTA); **f** hMrs2-lowEDTA (containing 1 mM EDTA, pH 8.0); **g** hMrs2-highEDTA (containing 5 mM EDTA, pH 6.8). Five protomers are displayed in different colors.

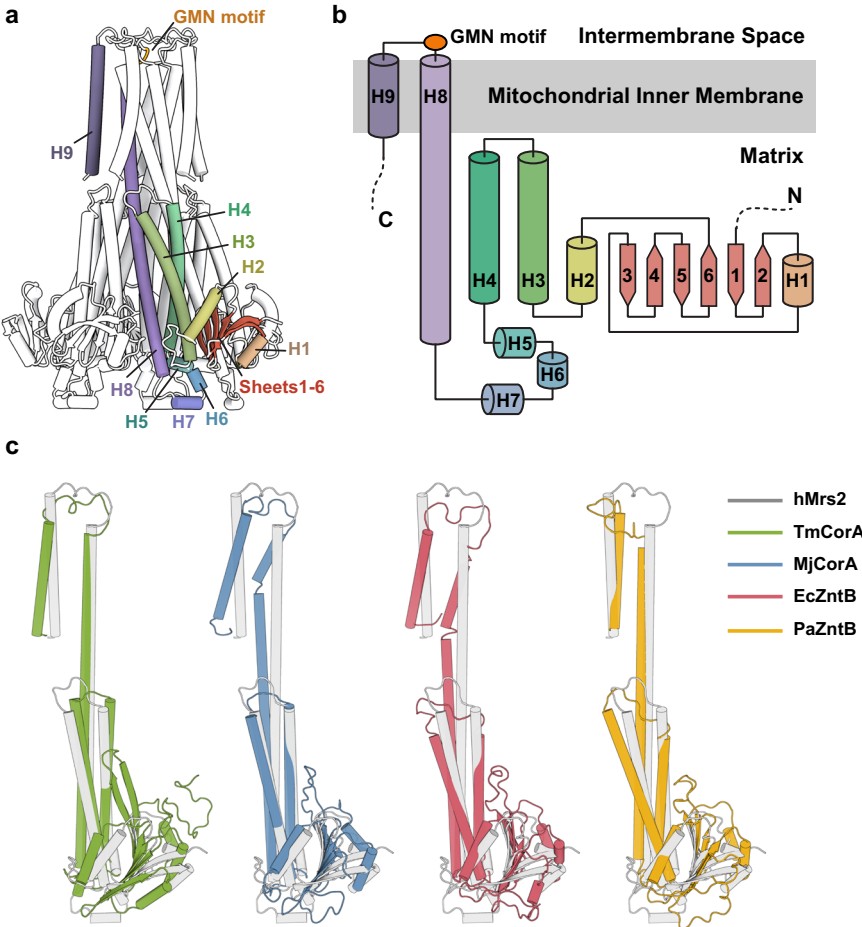

**Fig. 2 | Structure of the hMrs2 protomer. a** Illustration of the hMrs2 pentamer. One protomer is displayed in a different color while the other four protomers are shown in black and white. **b** Schematic diagram outlining the protein secondary structures of an hMrs2 protomer. **c** Comparisons of the hMrs2 protomer structures with those of other CorA family members: TmCorA (RCSB 4I0U; Cα RMSD 2.89 Å), MjCorA (RCSB 4EV6; Cα RMSD 2.77 Å), EcZntB (RCSB 5N9Y; Cα RMSD 3.17 Å) and PaZntB (RCSB 7NH9; Cα RMSD 3.01 Å).

aforementioned CorA structures were 3.84 Å for hMrs2 with TmCorA, 2.77 Å for hMrs2 with MjCorA, 3.17 Å for hMrs2 with EcZntB and 3.01 Å for hMrs2 with PaZntB.

## Channel assembly

Each hMrs2-Mg protomer consists of an N-terminal α/β domain with a central six-stranded mixed β sheets (β1-β6) flanked on one side by a short helix (H1) and on the other side by one short helix (H2) and two long helices (H3 and H4) (Fig. 2a, b). The two long helices form a helical bundle involving the matrix portion of the extra-long helix H8. Three short helices (H5, H6 and H7) are connected in a region between H4 and H8. The C-terminal part of helix H8 is the first transmembrane helix, and five of helices form the walls of the central ion-conduction pore. Downstream of helix H8, H9 forms a second transmembrane helix, and five H9 helices form the outer shell of the pore. The short loop H8–H9 is located within the mitochondrial intermembrane space and harbors the essential GMN motif. The general features of the hMrs2-Mg protomer are similar to those in TmCorA, MjCorA, EcZntB and PaZntB, which may indicate structural and functional conservation among different members of the CorA family, however two structural details of these proteins differ greatly (Fig. 2c and Supplementary Fig. 5). The first difference involves the N-terminal α/β domain, in which the number of β-strands and α-helices differ, resulting in a large difference in the protomer–protomer interface of the N-terminus. The second involves the orientation of H9 relative to H8, which results in a different position of the selection filter GMN motif.

The hMrs2 channel exhibits extensive intersubunit interactions, and most of these involve in the extra-long helix H8. From the intermembrane space side down to the matrix side, representative interactions between two neighboring protomers can be classified into seven major interfaces, which are shown in Supplementary Fig. 6. Due to the presence of the fivefold symmetric axis, all interactions between two adjacent protomers are repeated five times in the hMrs2 channel.

## Ligand recognition

In the hMrs2-Mg structure, four Mg$^{2+}$-binding sites and one Cl$^-$-binding site were identified (Fig. 3a). The first Mg$^{2+}$-binding site (called Mg-1) is located inside the pore mouth on the intermembrane side (Fig. 3a, b). Mg$^{2+}$ is coordinated by the classical GMN motif prevalent in the CorA family[29]. Mg-1 is at the center of the pentamer. From the side, Mg-1 appears to be an intermediate between the upper layer formed by five oxygen atoms of each residue N362 side chain and the lower layer formed by five carboxyl oxygen atoms in the main chain of each residue G360. The distance between all the oxygen atoms and Mg$^{2+}$ is approximately 4.3 Å, indicating that the coordinated Mg$^{2+}$ in the Mg-1 site is hydrated. The second Mg$^{2+}$-binding site (called Mg-2) is located in the pentameric center of the matrix part of the hMrs2 channel, positioned immediately below the inner leaflet of the membrane (Fig. 3a, c). Mg$^{2+}$ is coordinated by five carboxylic acid groups per D329 residue. The distance between the oxygen atom and Mg$^{2+}$ is approximately 4.4 Å, indicating that Mg$^{2+}$ at the Mg-2 site is hydrated. The third Mg$^{2+}$-binding site (called Mg-3) is located in the middle high region of the

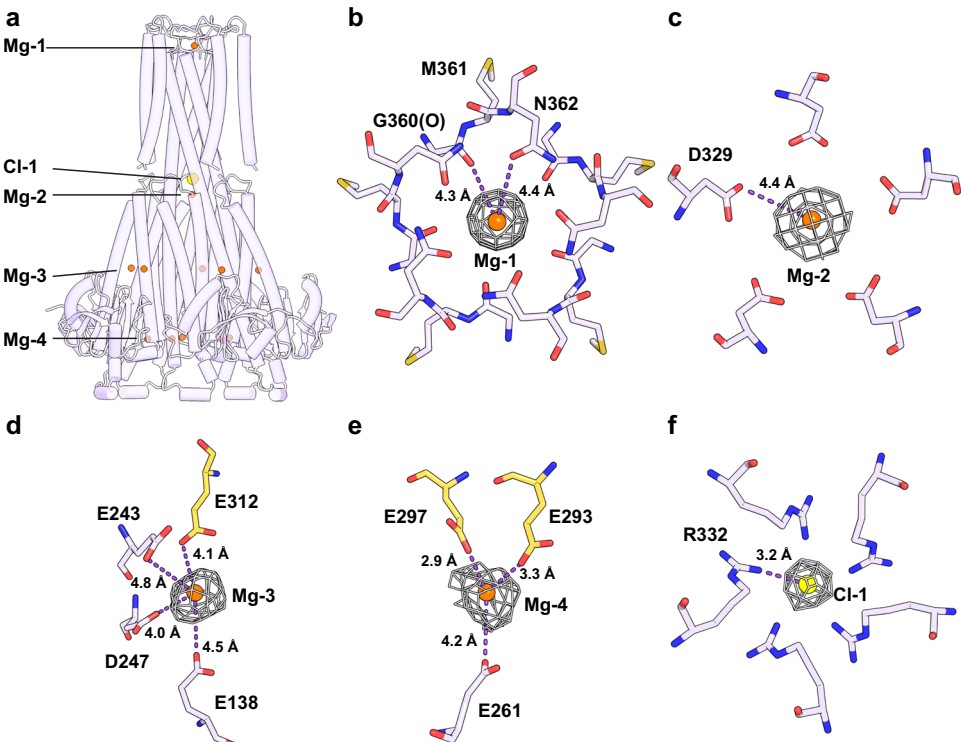

**Fig. 3 | Ligand recognition in the hMrs2-Mg structure. a** Overall picture of the ions bound in the hMrs2 pentamer structure. Ions appear as spheres of different colors. The ion-binding sites Mg-1 (**b**), Mg-2 (**c**), Mg-3 (**d**), Mg-4 (**e**), and Cl-1 (**f**). Mg$^{2+}$ ions are in orange, and Cl$^-$ ions are in yellow. The residues around different ions are shown as sticks. The gray mesh represents the density map of the ions, and the dotted line represents a salt bridge. The maps of Mg-1, Mg-2, Mg-3, Mg-4 and Cl-1 contoured at the 4.0 σ, 4.0 σ, 3.0 σ, 2.5 σ and 4.0 σ level, respectively. The distance between two atoms is indicated on the dotted line. Oxygen, nitrogen and carbon atoms are colored red, blue and light blue, respectively. The carbon atoms shown in (**d**) and (**e**) were from two protomers and therefore are presented differ in colors.

hMrs2 matrix part within the interface between two adjacent protomers (Fig. 3a, d). Mg$^{2+}$ is coordinated by three carboxylic acid residues (E138, E243 and D247) in one protomer and one carboxylic acid group of residue E312 in another protomer. The fourth Mg$^{2+}$-binding site (called Mg-4) is located in the near bottom region of hMrs2 within the interface between two protomers (Fig. 3a, e). Mg$^{2+}$ is coordinated at this site by two carboxylic acid residues (E297 and E293) in one protomer and one carboxylic acid group of residue E261 in another protomer. The Cl$^-$-binding site (called Cl-1) is located in the pentameric center of the transmembrane part of the hMrs2 channel, very close to the inner leaflet of the membrane (Fig. 3a, f). Cl-1 is directly above Mg-2. Cl$^-$ is coordinated by five guanidine groups per R322 (henceforth named the R-ring). The distance between the nitrogen atom and Cl$^-$ is approximately 3.2 Å. Of the five ligand-binding sites, three sites (Mg-1, Mg-2 and Cl-1) but not the other two binding sites (Mg-3 and Mg-4) are located in the three other aforementioned hMrs2 structures (hMrs2-rest, hMrs2-lowEDTA and hMrs2-highEDTA). Furthermore, Mg-1, Mg-2 and Cl-1 are unlikely to be misidentified due to artificial densities caused by the fivefold symmetry operation, as the density of each of the three sites is clearly visible in cryo-EM reconstruction of hMrs2-highEDTA without posing any symmetry (Supplementary Fig. 7).

Compared with ion-binding sites in the previously published structures (TmCorA, MjCorA, EcZntB and PaZntB), Mg-1, Mg-2, Mg-3 and Mg-4 are in similar positions to the ion-binding sites in the MjCorA structure[19]. In addition, Mg-1 and Mg-2 are in similar positions in the TmCorA structure[24]. Cl-1 has never been reported in CorA family proteins.

### Ion-conduction pathway

The surface map of the hMrs2 channel shows a long pore that extends from the intermembrane space to the matrix, forming an obvious ion-conduction pathway (Fig. 4a). The pore regions at both the intermembrane space end and the matrix space end are highly negatively charged. However, the pore region through the inner leaflet of the membrane is positively charged, which may act as a charge repulsion barrier for Mg$^{2+}$ conduction. The pore size distribution plot calculated with the HOLE software (2.2.005)[30] shows that the main contributor to this positively charged repulsion barrier is the R-ring, which forms the narrowest region (radius of approximately 1.1 Å) in the entire pore (Fig. 4b, c). As mentioned above, the R-ring forms site Cl-1, which accommodates one chloride ion. A pore with such a small diameter coupled with Cl$^-$ would likely prevent the conduction of ions. To test this hypothesis, MD simulations were carried out. Three replicas of the 500-ns simulations were performed (Fig. 4d, e and Supplementary Fig. 8), and the results showed that continuously distributed water channels were blocked by a chloride ion (Fig. 4d) and that they reformed after manual removal of the chloride ion (Fig. 4e). We also calculated the potential of mean force (PMF) to quantify the main energy barrier of hydrated Mg$^{2+}$ ions passing through the hMrs2 channel and compared the energy curves in the presence or absence of Cl$^-$. The major energy barrier in both curves was at the same location representing the point where the hydrated Mg$^{2+}$ passes through site Cl-1 (Fig. 4f and Supplementary Fig. 8). As the hydrated Mg$^{2+}$ flows into the matrix, the free energy gradually decreases. Unexpectedly, the Cl$^-$ significantly reduced the free energy, thereby favoring Mg$^{2+}$ permeation (Fig. 4f). These results suggest that Cl$^-$ plays a paradoxical role as both a barrier to and a facilitator of Mg$^{2+}$ permeation.

### Mg$^{2+}$ permeation

To address the molecular mechanism of Mg$^{2+}$ permeation in hMrs2 channels, we carried out a mitochondrial Mg$^{2+}$ uptake assay. Using purified mitochondria from HEK-293F cells, endogenous hMrs2 took up Mg$^{2+}$ when Mg$^{2+}$ was not removed from the matrix (Fig. 5a). Overexpression of hMrs2 in cells greatly increased the protein levels

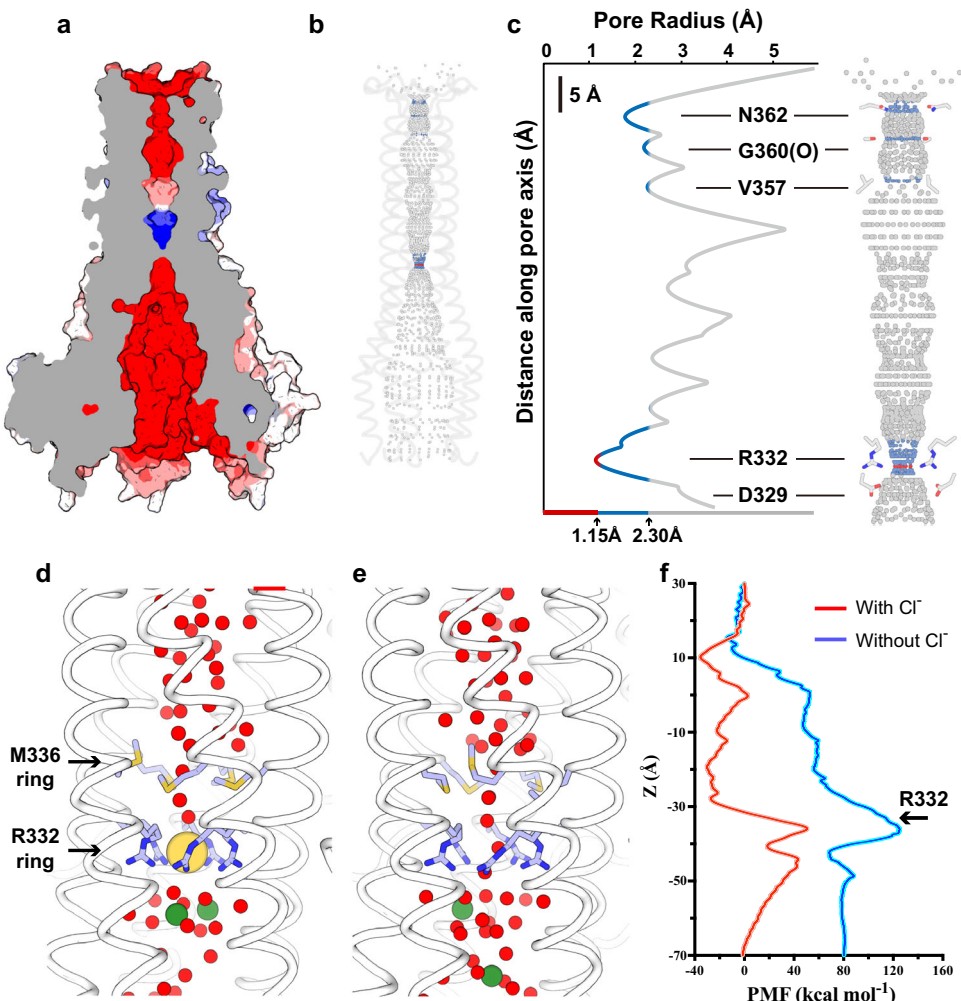

**Fig. 4 | Ion-conduction pathway. a** Surface potential map of the hMrs2 pentamer. The hMrs2 pentamer in the surface representation is colored according to electrostatic surface potentials from −10 to +10 kT e$^{-1}$ (from red to blue). **b** Ion conduction pore inside the hMrs2 pentamer. Cartoon representations of two face-to-face protomers are shown. Blue, red, and gray dots represent the path of the pore. Red indicates a pore size <1.15 Å; blue indicates a pore size ranging from 1.15 to 2.3 Å; and gray indicates a pore size >2.3 Å. **c** Pore radius profile along the central axis (left) and the corresponding ion-conduction pathway (right). The side chains of

five residues (N362, G360, V357, R332 and D329) are presented as sticks. Representative snapshot of a continuous water channel with Cl$^-$ (**d**) or without Cl$^-$ (**e**) in the hMrs2 pentamer in a 500-ns MD simulation. This simulation was repeated 3 times. Red, yellow and green spheres represent water molecules, Cl$^-$ and Na$^+$, respectively. The illustration of hMrs2 is shown in light blue. **f** PMF profiles along the ion-conduction pathway with or without Cl$^-$. The uncertainty error is calculated based on Monte Carlo bootstrapping and shown in colored shadows (almost unnoticeable).

in mitochondria (Supplementary Fig. 9), largely increasing the amount of mitochondrial Mg$^{2+}$ taken up (Fig. 5a). To investigate the importance of Cl$^-$ binding, we replaced Arg332 with an alanine residue (R332A), lysine residue (R332K) or glutamate residue (R332E) and then measured mitochondrial Mg$^{2+}$ uptake. The amount of Mg$^{2+}$ permeating the mitochondria in cells overexpressing any of these three mutants was largely unchanged (Fig. 5a), suggesting that Cl$^-$ plays a regulatory, not a decisive role, in Mg$^{2+}$ permeation. Further experiments will be required to determine the exact role of Cl$^-$ ion. We further examined the effect of inner mitochondrial membrane potential on Mg$^{2+}$ permeation. Purified mitochondria treated with carbonyl cyanide 4-(trifluoromethoxy) phenylhydrazone (FCCP) or valinomycin reduced the amount of Mg$^{2+}$ taken up (Fig. 5b). In particular, mitochondrial overexpression of hMrs2 profoundly attenuated Mg$^{2+}$ uptake after treatment with FCCP or valinomycin. These results suggested that Mg$^{2+}$ permeation via hMrs2 is highly dependent on the membrane potential. Our results correspond to previous results indicating that the uptake of mitochondrial Mg$^{2+}$ by yeast Mrs2p is driven by the mitochondrial membrane potential (ΔΨ)[10].

To further understand how hMrs2 enables Mg$^{2+}$ membrane permeation, we used steered-MD simulation to employ water coordination during Mg$^{2+}$ permeation. In bulk solution, the first hydration shell of Mg$^{2+}$ coordinated 6 water molecules, which was consistent with previous studies[31,32]. Throughout the pore, the number of water molecules coordinated by Mg$^{2+}$ was between 4 and 6, indicating that Mg$^{2+}$ is partially dehydrated during permeation (Fig. 5c). The water coordination profile of the entire pore has two depths. The first was found at the entrance of the pore consisting of five Asn362 residues, and the second was found at the positive charge repulsion barrier formed by the R-ring. The number of Mg$^{2+}$ coordination water molecules in both places decreased from 6 to 4. When passing through the R-ring, the Mg$^{2+}$ hexahydrate unexpectedly attracted the Cl$^-$ bound in the R-ring, and this force moved Cl$^-$ in the direction of the mitochondrial intermembrane space (Fig. 5d). Subsequently, Mg$^{2+}$ formed a complex with Cl$^-$, which may explain the reduction in the coordination water number from 6 to 4 (Fig. 5d). The coordination of Cl$^-$ protected Mg$^{2+}$ from positive electrostatic repulsion caused by the R-ring, thereby promoting extrusion of the complex [Mg$^{2+}$·Cl$^-$·4H$_2$O] through the R-ring. Facilitated by the negatively charged residue D329, the

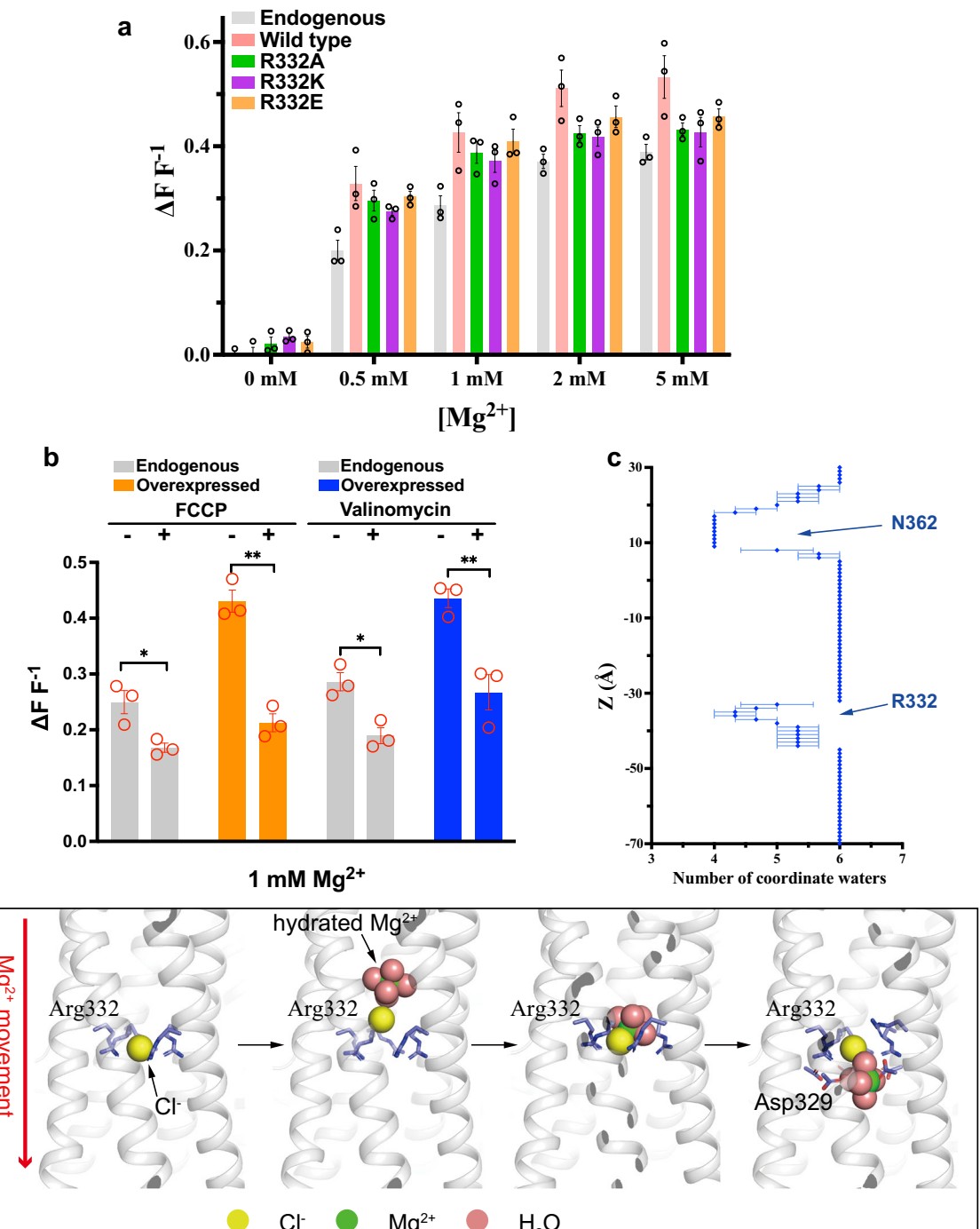

**Fig. 5 | Cl⁻ assists Mg²⁺ permeation. a** Relative Mg²⁺ uptake capacity of purified mitochondria overexpressing wild-type hMrs2 (denoted as Wild type) or mutant hMrs2 (denoted as R332A, R332K and R332E) was quantified and shown in a bar graph. Mitochondria with endogenous hMrs2 (denoted as Endogenous) served as controls. Individual data points are shown as black circles. Error bars denote the standard error of the mean. $n = 3$ independent experiment. **b** Relative Mg²⁺ uptake capacity of purified mitochondria overexpressing wild-type hMrs2 or endogenous hMrs2 after treatment with FCCP or valinomycin. Individual data points are shown as red circles. Error bars denote the standard error of the mean. $n = 3$ independent experiment. An unpaired two-tailed $t$-test was used to analyze data. * $P < 0.05$, ** $P < 0.01$. $P$-values are as follows: endogenous hMrs2 without vs with FCCP treatment, $p = 0.02148$; overexpressed hMrs2 without vs with FCCP treatment,

$p = 0.00104$; endogenous hMrs2 without vs with valinomycin treatment, $p = 0.011$; overexpressed hMrs2 without vs with valinomycin treatment, $p = 0.00923$. **c** The profile showing that Mg²⁺ coordinates the water number distribution along the ion-conduction pathway. Error bars denote the standard error of the mean. $n = 3$ independent experiment. **d** Cl⁻ facilitates Mg²⁺ movement through the R-ring. The R-ring consists of five Arg332 residues bound to Cl⁻ (far left). When fully hydrated Mg²⁺ approaches the R-ring, Cl⁻ moves toward Mg²⁺, forming a complex, and Mg²⁺ is partially dehydrated (left). The complex is captured by the R-ring (right). The complex squeezes through the R-ring and the components separate from each other (far right). Five H8 helices are shown in black and white. Oxygen, nitrogen and carbon atoms are colored red, blue and light blue, respectively.

4-hydrated $Mg^{2+}$ dissociated from $Cl^-$, was rehydrated with water, and moved toward the matrix (Fig. 5d).

## Discussion

In this study, we report four structures of the hMrs2 channel, a eukaryotic member of the CorA family, under different conditions. Similar to other members of the CorA family, hMrs2 is assembled as a holo-pentamer. Each protomer harbors two transmembrane helices and a conserved GMN motif, i.e., an ion-selective filter. Four $Mg^{2+}$-binding sites were identified in the hMrs2-Mg structure. Their locations were similar to the ion-binding sites in the published CorA structures[19]. A possible $Cl^-$-binding site, composed of five Arg332 residues (R-ring), was identified in hMrs2. Sequence alignment analysis indicated that the residue Arg332 was not conserved among members of the CorA family (Supplementary Fig. 5), implying that hMrs2 might employ a distinct $Mg^{2+}$ permeation mechanism compared to TmCorA[18,20,24,26].

Previous studies have suggested that $Mg^{2+}$ binding sites in the TmCorA intracellular domains may function as switches that regulate TmCorA opening[15–17]. Structural studies have shown that the loss of $Mg^{2+}$ causes TmCorA to undergo a change from a symmetric pentamer structure (closed state) to a pronounced asymmetric structure (open state)[18,33]. However, at different pH values and after EDTA treatments at different concentrations, all four hMrs2 structures formed symmetric holo-pentamers. Similar results have been previously reported for EcZntB, which maintained a symmetric pentameric state even after EDTA treatment[27]. Moreover, the four hMrs2 structures were very similar in terms of the pentamer and protomer structures. These findings suggest that no significant structural rearrangement occurs when the hMrs2 loses $Mg^{2+}$ binding (hMrs2-Mg versus hMrs2-high-EDTA) in the matrix domain. In MD simulations after $Cl^-$ was manually removed from the pore, the continuous distribution of water channels was observed in the ion-conduction pore without the need for the conformational change in the main chain of the hMrs2 structure. In addition, when the mitochondrial matrix was maintained at physiological $Mg^{2+}$ concentrations (-0.8 mM), purified mitochondria took up large amounts of $Mg^{2+}$ when they overexpressed the hMrs2 protein. Taken together, the matrix $Mg^{2+}$ concentration appears to be dispensable for $Mg^{2+}$ permeation via hMrs2, reinforcing the argument that hMrs2 and TmCorA may employ different $Mg^{2+}$ permeation mechanisms.

Previous reports showed that purified yeast mitochondria overexpressing Mrs2p after treatment with valinomycin showed reduced $Mg^{2+}$ uptake[10]. For TmCorA, the membrane potential facilitates the uptake of magnesium ions, as shown via liposome reconstitution and cellular uptake assays[23,34]. Our results showed that FCCP or valinomycin treatment greatly reduced the amount of $Mg^{2+}$ taken up by mitochondria overexpressing hMrs2, suggesting that the membrane potential is the main driver of $Mg^{2+}$ permeation via hMrs2. We also observed that mitochondria with endogenous hMrs2 took up $Mg^{2+}$ after treatment with FCCP or valinomycin. We suspect that another unidentified mitochondrial $Mg^{2+}$ transporter may function independent of the membrane potential.

Considering the results of our structural and MD simulations and functional studies, we propose a model for $Mg^{2+}$ permeation in the hMrs2 channel. First, an increase in cytosolic $Mg^{2+}$ concentration causes $Mg^{2+}$ to enter the selection filter, GMN motif. Next, the membrane potential drives $Mg^{2+}$ flow to the matrix. The R-ring is then used as a charge repulsion gate to impede the forward movement of $Mg^{2+}$. Finally, $Cl^-$ may play a ferry-like role in facilitating $Mg^{2+}$ transport into the matrix through the R-ring. The principle of "anion-assisted cation permeation" has also been observed in the calcium channel Orai protein located in the plasma membrane[35,36].

It is worth noting that while the R-ring in the transmembrane pore region of the hMrs2 channel is not present in other members of the CorA family, wider arginine rings in the matrix domain have been observed in the structures of TmCorA, MjCorA, EcZntB, and PaZntB[19,22,27,28]. Consequently, the possibility of chloride-assisted $Mg^{2+}$ transport cannot be entirely disregarded for other members of the CorA family, at least for the current time being.

## Methods

### Plasmid construct

The gene encoding *Homo sapiens* Mrs2 (UniProt ID: Q9HD23) was synthesized with a C-terminal FLAG tag (GENEWIZ, Inc.) and cloned into a pcDNA4/Myc-His A vector (Thermo Fisher). The hMrs2 mutants (R332A, R332E, and R332K) were generated through standard PCR-based mutagenesis methods and verified by DNA sequencing (GENE-WIZ). Primers sequences were listed in Supplementary Data 1. The expression plasmids were amplified in *E. coli* and purified with a GoldHi EndoFree Plasmid Maxi Kit (CWBio).

### Protein expression and purification

HEK-293F cells were cultured in SMM293-TII expression medium (Sino Biological), supplemented with 5% $CO_2$ at 37 °C. Prior to transfection, the cells were diluted to a density of $0.7 \times 10^6$ cells $mL^{-1}$ and cultured for 24 h. Transient transfection was performed using PEI 25k (Polysciences, Inc). For $1.0 \times 10^9$ cells, 1.0 mg of plasmid and 3 mg of PEI were preincubated in 90 mL of fresh medium for 20 min. Then, the DNA/PEI mixture was added to the cell cultures. After another 24 h, SMS 293-SUPI cell culture supplement (Sino Biological) was added to the cultures at a ratio of 3.4% (v $v^{-1}$). The cells were harvested 60 h after transfection and stored at −80 °C.

For protein purification, each liter of cells was resuspended in 100 mL of buffer consisting of 20 mM Tris-HCl pH 8.0, 200 mM NaCl, 1% (w $v^{-1}$) n-dodecyl-ß-D-maltoside (DDM, Anatrace) and 1 × cocktail (TargetMol). Cell lysates were incubated at 4 °C for 2 h, followed by ultracentrifugation at $120,000 \times g$ for 45 min. The resulting supernatant was then incubated with Anti-DYKDDDDK Tag (L5) Affinity Gel (BioLegend) at 4 °C for 4 h. The resin was washed sequentially with buffer A (20 mM Tris-HCl pH 8.0, 200 mM NaCl and 0.03% (w $v^{-1}$) DDM), buffer B (20 mM Tris-HCl pH 8.0, 200 mM NaCl, 0.03% DDM, 1 mM $MgCl_2$ and 10 mM ATP), and buffer C (20 mM Tris-HCl pH 8.0, 200 mM NaCl and 0.007% (w $v^{-1}$) GDN (Anatrace)). The target protein was eluted with buffer C containing 0.4 mg $mL^{-1}$ DYKDDDDK peptide. The eluent was concentrated to a 1 mL volume using a Microsep Advance ultrafiltration device with a 100-kDa cutoff (Pall Life Sciences). Subsequently, the concentrated eluent was loaded onto a Superose 6 increase 10/300 GL column (Cytiva), which had been pre-equilibrated with buffer D (20 mM Tris-HCl pH 8.0, 150 mM NaCl, 1 mM DTT and 0.007% (w $v^{-1}$) GDN). For the hMrs2-Mg structure, 20 mM $MgCl_2$ was added to buffer D. For the hMrs2-lowEDTA structure, 1 mM EDTA was added to buffer D. For the hMrs2-highEDTA structure, buffer E (10 mM MES pH 6.8, 150 mM NaCl, 5 mM EDTA, 0.5 mM EGTA, 1 mM DTT and 0.007% (w $v^{-1}$) GDN) was used for gel filtration. The peak fractions were pooled and concentrated to a final concentration of 5–15 mg $mL^{-1}$ for cryo-EM.

### Cryo-EM sample preparation and imaging

An aliquot of 2.5 µL purified hMrs2 was applied to glow-discharged Au 200 mesh R2/1 holey carbon grids (QUANTIFOIL). The grids were blotted for 4 s with blot force 7–8 at 8 °C and 100% humidity. Subsequently, the grids were rapidly plunge-frozen in liquid ethane cooled by liquid nitrogen using the FEI Vitrobot Mark IV.

A total of 690 image stacks were collected for hMrs2-rest sample by a 300 kV Titan Krios G3 cryo-electron microscope (FEI) equipped with K2 Summit direct-electron detector (Gatan) using SerialEM software (3.8)[37] in counting mode with the pixel size of 1.014 Å and defocus range of −0.8 to −1.6 µm. Each movie stack was dose-fractionated over 40 frames by 8 s exposure with a dose rate of 1.35 $e^-$ per $Å^2$ per frame. hMrs2-Mg, hMrs2-lowEDTA and hMrs2-highEDTA samples were

automatically imaged with EPU software (2.14) in a 300 kV Titan Krios G3 cryo-electron microscope (FEI) equipped with Falcon 4 direct-electron detector and Selectris energy filter (ThermoFisher). Image stacks were recorded in EER mode, with a pixel size of 0.93 Å and a defocus range of −0.8 to −1.6 μm. The image stacks in EER format were converted to TIFF format using the relion_convert_to_tiff command in RELION software (4)[38]. For hMrs2-Mg samples, a total of 3329 image stacks were collected with a total dose of 51.39 e$^-$ Å$^{-2}$, each compressed image stack containing 45 frames with dose rate of 1.13 e$^-$ per Å$^2$ per frame. For hMrs2-lowEDTA samples, a total of 3217 image stacks were collected with a total dose of 51.39 e$^-$ Å$^{-2}$, each compressed image stack containing 45 frames with dose rate of 1.13 e$^-$ Å$^{-2}$ per frame. For hMrs2-highEDTA samples, a total of 1708 image stacks were collected with a total dose of 59.97 e$^-$ Å$^{-2}$, each compressed image stack containing 45 frames with dose rate of 1.33 e$^-$ Å$^{-2}$ per frame.

## Image processing

Image stacks were processed by MotionCor2 software (1.3.2)[39] for 5 × 5 patch drift correction with dose weighting. Non-dose-weighted images were used for contrast transfer function (CTF) estimation by CTFFIND software (4.0.8)[40]. Poor quality images were manually removed before the particles were picked. The particles were picked up semi-automatically by Gautomatch software (0.56) (www.mrc-lmb.cam.ac.uk/kzhang/) and extracted by RELION software (3.1)[41] with a 260-pixel box size.

For the hMrs2-Mg dataset, 473,653 particles picked from 2,375 selected images were subjected to 2 rounds of 2D classification in RELION software (4) to remove bad particles, yielding 147,386 particles. The selected particles were then used to generate an initial model by the Ab initio program in CryoSPARC software (3.3.2)[42]. The initial model was pass-filtered to 30 Å resolution and applied for 3D classification in RELION software (4). After two rounds of 3D classification, the selected subset containing 60,865 particles was used for 3D auto-refine and Bayesian polishing with C5 symmetry. The polished particles were imported into CryoSPARC software (3.3.2), followed by non-uniform refinement, CTF refinement, and symmetry expansion according to C5 symmetry. The mask output at the non-uniform refinement step was used as the initial mask for the subsequent local refinement step. The local refinement procedure, integrating a dynamically generated mask in each refinement iteration, was executed in conjunction with non-uniform adaptive regulation, leading to the production of a final map with a resolution of 2.6 Å with C1 symmetry. The automatically sharpened map derived from the local refinement step was subsequently employed for subsequent model building and refinement.

For the hMrs2-rest dataset, 170,290 particles picked from 635 selected images were subjected to 2 rounds of 2D classification in RELION software (3.1) to remove bad particles, yielding 36,283 particles. The selected particles were then used to generate an initial model by the Ab initio program in CryoSPARC software (3.3.2). The initial model was pass-filtered to 30 Å resolution and applied for 3D classification in RELION software (3.1). After two rounds of 3D classification, the selected subset containing 20,800 particles was used for 3D auto-refine and Bayesian polishing with C5 symmetry. The polished particles were then imported into CryoSPARC software (3.3.2) for homogeneous refinement and then symmetry expansion according to C5 symmetry. The mask output at the homogeneous refinement step was used as the initial mask for the subsequent Local Refinement step. The Local Refinement procedure, integrating a dynamically generated mask in each refinement iteration, was executed in conjunction with non-uniform adaptive regulation, leading to the production of a final map with a resolution of 2.9 Å with C1 symmetry. The automatically sharpened map derived from the Local Refinement step was subsequently employed for the subsequent model building and refinement.

For the hMrs2-highEDTA dataset, 374,846 particles picked from 1244 selected images were subjected to 2 rounds of 2D classification in RELION software (4)[41] to remove bad particles, yielding 58,010 particles. The selected particles were then used to generate an initial model by the Ab initio program in CryoSPARC software (3.3.2). The initial model was pass-filtered to 30 Å resolution and applied for 3D classification in RELION software (4). After a round of 3D classification, the selected subset containing 42,527 was used for 3D auto-refine and Bayesian polishing with C5 symmetry and then imported into CryoSPARC software (3.3.2). After a further round of 2D classification, 34,277 particles were selected and subsequently subjected to non-uniform refinement, CTF refinement, and then symmetry expansion according to C5 symmetry. The mask output at the non-uniform refinement step was used as the initial mask for the subsequent Local Refinement step. The Local Refinement procedure, integrating a dynamically generated mask in each refinement iteration, was executed in conjunction with non-uniform adaptive regulation, leading to the production of a final map with a resolution of 2.7 Å with C1 symmetry. The automatically sharpened map derived from the Local Refinement step was subsequently employed for the subsequent model building and refinement.

For the hMrs2-lowEDTA dataset, 631,480 particles picked from 2324 selected images were subjected to 2 rounds of 2D classification in RELION software (4) to remove bad particles, yielding 203,681 particles. The selected particles were then used to generate an initial model by the Ab initio program in CryoSPARC software (3.3.2). The initial model was pass-filtered to 30 Å resolution and applied for 3D classification in RELION software (4). After two rounds of 3D classification, the selected subset containing 37,000 particles was used for 3D auto-refine and Bayesian polishing with C5 symmetry. The polished particles were imported into CryoSPARC software (3.3.2) and subsequently subjected to non-uniform refinement, CTF refinement, and then symmetry expansion according to C5 symmetry. The mask output at the non-uniform refinement step was used as the initial mask for the subsequent Local Refinement step. The Local Refinement procedure, integrating a dynamically generated mask in each refinement iteration, was executed in conjunction with non-uniform adaptive regulation, leading to the production of a final map with a resolution of 2.5 Å with C1 symmetry. The automatically sharpened map derived from the Local Refinement step was subsequently employed for the subsequent model building and refinement.

The resolution of the maps was evaluated using the "gold-standard" FSC = 0.143 criterion. The local resolution was calculated using ResMap software (1.1.4)[43]. The workflows for the data collection and reconstruction are shown in Supplementary Figs. 1–4. All data collection and reconstruction statistics are presented in Supplementary Table 1.

## Model building and refinement

The hMRS2-rest model building used a predicted human Mrs2 (Uni-Prot ID: Q9HD23) model from the Alphafold2 database. The predicted coordinates were fitted into hMRS2-rest map in UCSF ChimeraX (1.5)[44], and pre-fitted single-chain model was refined by PHENIX software (1.19)[45] in the real_space_refinement with model morphing and secondary structure and geometry restraints. After the real space refinement, the regions with poor density fitting and side-chain orientation were manually built and refined according to the density map in COOT software (0.8.9.3)[46]. After five chains of the model were manually fitted and refined, the entire model was refined by PHENIX software (1.19) real_space_refinement with secondary structure and geometry restraints. Based on refined hMRS2-rest model, hMRS2-Mg, hMRS2-lowEDTA and hMRS2-highEDTA structures were built and refined by COOT software (0.8.9.3) and PHENIX software (1.19). The quality of the models was assessed by MolProbity module of PHENIX software (1.19). Refinement statistics and validation statistics for hMRS2-rest, hMRS2-

Mg, hMRS2-lowEDTA and hMRS2-highEDTA models are shown in Supplementary Table 1.

## Mitochondrial isolation and $Mg^{2+}$ uptake assay

Mitochondrial isolation experiments were performed in accordance with the literature[14]. HEK-293F cells were harvested by centrifugation ($600 \times g$, 5 min at 4 °C). The cells were then washed with ice-cold PBS and resuspended in cold isolation buffer (IB; 5 mM HEPES-KOH, 70 mM sucrose, 210 mM mannitol, pH 7.2) at a ratio of 4 mL per gram of cells. BSA was added to the solution so that the final concentration was 0.25%. The resuspension was added to a glass homogenizer, and the pestle was pumped up and down 10-15 times. An additional threefold volume of buffer IB was added to the homogenate, and the pellet was removed after two rounds of centrifugation ($1000 \times g$, 10 min at 4 °C each time). The supernatant was then centrifuged ($10,000 \times g$, 10 min, 4 °C) to obtain mitochondria. Purified mitochondria were quantified by Bradford's method.

$Mg^{2+}$ uptake assay buffer (AB) consisted of IB supplemented with 0.5 mM ATP, 0.2% succinate, and 0.01% pyruvate. Mitochondria were resuspended at 0.5 mg mL$^{-1}$ in AB and loaded with 2 μM Mag-fluo-4 (Invitrogen) and 0.02% Pluronic F-127 (Thermo Fisher) for 25 min at room temperature (RT). Mitochondria were washed twice with AB and then incubated for 35 min. The mitochondria were then washed twice more with AB and dispensed into a 96-well plate at a ratio of 96 μL per well. Fluorescence was monitored at RT with Spark multimode microplate reader (Tecan) using filter sets (excitation 485 nm, emission 520 nm) before and after $Mg^{2+}$ injection. The average fluorescence value before and after $Mg^{2+}$ addition was subtracted and plotted in a bar graph.

## Western blotting

The primary antibodies: rabbit anti-MRS2 antibody (Sigma–Aldrich, HPA017642, 1:1000) and mouse anti-cytochrome C (Abcam; ab110325, 1:2000), and the secondary antibodies: rabbit anti-mouse IgG (Abcam, ab6728, 1:8000) and goat anti-rabbit antibody (Abcam, ab6721, 1:2000) were diluted in TBST. Samples were loaded on 12.5% SDS–PAGE gels, and the proteins were transferred to PVDF membranes. The membranes were incubated with 5% nonfat dry milk for 1 h at RT. After washing 3 times with TBST buffer, primary antibodies were incubated with the membranes for 1 h at RT. After washing another 3 times with TBST, the secondary antibodies were incubated with the membrane for 1 h at RT. Finally, after washing 3 times with TBST, the membranes were soaked in a chemiluminescent HRP substrate (Millipore) and imaged on a Tanon 5200 Multi Chemiluminescent Imaging System (Tanon).

## Molecular dynamics (MD) simulations

The hMrs2-Mg structure was prepared for the MD simulations. The bound chlorine ion (Cl$^-$) and magnesium ion ($Mg^{2+}$) were retained. The protonation state for titratable residues was determined using the H++ program[47]. All proteins were capped at the N- and C-termini with acetyl and methyl amide groups, respectively. All missing side-chain non-hydrogen atoms and hydrogen atoms were added automatically using the LEaP module in AMBER-2020 (http://ambermd.org/AmberMD.php). The hMrs2 protein was inserted into 135 Å × 135 Å bilayers of palmitoyl oleoyl phosphatidyl choline (POPC) with the pore axis aligned parallel to the z axis by using the VMD software (1.9.3) Membrane Builder plugin[48]. The protein-POPC system was then solvated in SPC/E water boxes[49] and neutralized by 0.15 M NaCl and 0.02 M MgCl$_2$. The size of the system Mrs2 with Cl$^-$ (Mrs2-withCl) was initially measured as 135 Å × 135 Å × 180 Å. The Cl$^-$ from the Mrs2-withCl system was omitted to build the Mrs2 without Cl$^-$ (Mrs2-noCl) system. One sodium ion was removed randomly to neutralized the system. The systems were consisted of 316,626 and 316,624 atoms, respectively (Supplementary Table 2).

All MD simulations were performed with AMBER-2020 with the PMEMD engine (http://ambermd.org/AmberMD.php) with the AMBER force field FF19SB[50], LIPID21[51], and SPC/E water model[49,52]. The Na$^+$ and $Mg^{2+}$ ions parameters were provided by the in-silico studies (Supplementary Table 3), in which the hydration free energy and hydrated radius of each ion were computed and fitted to experimental quantities in bulk solution[31,53]. These parameters had been used successfully to investigate the hydrated effect of several cations like $Mg^{2+}$ in nanopores or nanochannels[54,55]. The cut-off for non-bonded interactions was set to 12. The covalent bonds involving hydrogen atoms were constrained by the SHAKE algorithm[56]. The long-range electrostatic interactions were treated by the Particle Mesh Ewald (PME) algorithm[57]. The 10,000 steps minimization was carried out for each system, followed by the thermalization of each system heating from 0 K to 310 K in 500 ps using the Langevin thermostat[58]. The lipids were equilibrated for 30 ns with the proteins constrained (with force constant 50 kcal mol$^{-1}$ Å$^{-2}$), followed by the 20-ns optimization for missing atoms with the other protein atoms constrained with the gradually reduced harmonic restraints (50, 5 and 0.5 kcal mol$^{-1}$ Å$^{-2}$ for 16, 2 and 2 ns of simulation time, respectively). After that, the 20-ns equilibration phase was carried out, without any constraint applied for the entire system. Finally, three independent 500-ns production phase was performed for each system. The NPT ensemble was utilized at a constant pressure of 1 bar by the Berendsen barostat[59]. The time step was set to 2 fs. The frames were saved every 5000 steps for analysis.

## Potential of mean force (PMF) calculations

Due to the size (<2 Å) of two narrow regions (R332 and N362) of the hMrs2 pore is smaller than the size (approximately 2.4 Å) of the first hydration shell of magnesium ion[60], a high energy barrier exists as the $Mg^{2+}$ passes through the pore. The steered-MD simulations[61] for the Mrs2-withCl and Mrs2-noCl system were carried out to study the free energy change during $Mg^{2+}$ permeation. The last snapshot from one selected 500-ns MD trajectory was used as the starting point of the steered-MD for each system. To produce the steered-MD trajectories, the $Mg^{2+}$ position along the z direction (vertical to the membrane plane) was set to the reaction coordinate (RC) after placing the center of the lipids into the origin and a harmonic restraint (2.5 kcal·mol$^{-1}$·Å$^{-2}$) was used to steer the $Mg^{2+}$ permeation through the Mrs2 from intermembrane space to the matrix. The profiles of the coordinated water number were calculated based on three independent steered-MD simulations.

An umbrella sampling procedure[62] was used in 51 windows which starts from the extracted frames from steered-MD trajectory and dispersed uniformly along the RC where the distance ranges from −70 Å to +30 Å, at 2 Å intervals. In each simulation, the ion position along the Z direction was restrained using a harmonic restraint with a force constant 2.5 kcal mol$^{-1}$ Å$^{-2}$. The PMF was generated by fully sampling the configuration space in each window and then continued until the convergence of PMF was achieved. The total accumulated simulation time of trajectories used for the PMF calculations is 1.02 μs for Mrs2-withCl and Mrs2-noCl system, respectively. The PMF profiles utilized the WHAM software (2.0.11) (http://membrane.urmc.rochester.edu/wordpress/?page_id=126) and used the bin width of 0.1 Å.

## Quantification and statistical analysis

The number of independent experiments (N) and the relevant statistical parameters for each experiment (such as mean or standard deviation) are described in the pertinent figure legends. No statistical methods were used to pre-determine sample sizes. An unpaired two tailed $t$-test was performed to determine statistical significance of effects. Differences with a $P$-value of <0.05 were considered significant: *$P < 0.05$ and **$P < 0.01$.

**Reporting summary**

Further information on research design is available in the Nature Portfolio Reporting Summary linked to this article.

## Data availability

The data that support this study are available from the corresponding authors upon request. Cryo-EM maps have been deposited in the Electron Microscopy Data Bank (EMDB) under accession codes EMD-35630 (hMrs2-Mg), EMD-35633 (hMrs2-rest), EMD-34632 (hMrs2-low-EDTA), and EMD-35631 (hMrs2-highEDTA). The coordinates have been deposited in the Protein Data Bank (PDB) under accession codes 8IP3 (hMrs2-Mg), 8IP6 (hMrs2-rest), 8IP5 (hMrs2-lowEDTA), and 8IP4 (hMrs2-highEDTA). Previously solves structures were accessed from the PDB using 4I0U, 4EV6, 5N9Y and 7NH9. The MD simulations data have been deposited to Zenodo [https://doi.org/10.5281/zenodo.8153491]. Source data are provided with this paper.

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

## Acknowledgements

We are grateful to thank the technical assistance in the Center of Cryo-Electron Microscopy (CCEM), Zhejiang University on Cryo-EM for data acquisition and Dr. J. Lu for the support of electron microscopy at Nankai University. This work was supported by National Natural Science Foundation of China (grant 32271288 to X.Y.; grant 32071231 to Y.S.), Innovation Talent Program by Haihe Laboratory of Synthetic Biology (grant 22HHSWSS00009 to X.Y), Fundamental Research Funds for the Central Universities (grants 63231199 and 63223039 to X.Y.) and National Key Research and Development Program of China (2018YFA0507700 to X.Z.).

## Author contributions

M.L. did the protein expression, purification, sample preparation, cryo-EM data collection and Mg2+ uptake assay; Yang Li performed the MD simulations; Y.R and T.W. did the protein expression and purification; Y.W. performed the electrophysiology experiment; Yue Lu, J.L., X.L., S.C., X.Z. and X.Y. did cryo-EM data collection; X.Y. conducted the cryo-EM reconstruction; M.L., Yang Li, X.Y. and Y.S. analyzed the data, designed the study and wrote the paper. All authors discussed the results and contributed to the manuscript preparation.

## Competing interests

The authors declare no competing interests.
