## [Peer Review File · Nature Communications]

Molecular basis of Mg²⁺ permeation through the human mitochondrial Mrs2 channelReviewers' Comments:

Reviewer #1:

Remarks to the Author:

In this manuscript Li et al., report the long-awaited structure of Mrs2 channel from CorA superfamily of proteins. This a great milestone for the field, and there are some interesting observations like the stability of the protein even after extensive EDTA treatment and the putative involvement of chloride ions. Nevertheless I have some concerns and suggestions to the authors, which can hopefully improve their manuscript to be considered for publication in Nature Communications.

Major issues:

The comparison of the obtained Mrs2 structures with the known structures should be extended also to CmaX protein doi: 10.1016/j.ijbiomac.2021.06.130 as it also has somewhat different geometry. It is advisable to provide at least rmsd values in the analysis.

The authors claim the unique role of arginines and chlorides - but this is not novel. This was already discussed back in 2009 by Tan et al., DOI: 10.1002/pro.215

where one of their main conclusions is that "the transportation of metal ions can be further enhanced by binding chloride ions that can significantly lower the electrostatic barrier across the pore".

Furthermore despite indeed the Arginine which the authors propose to bind chloride ions is not conserved in the sequence, most likely it has at least some structural conservation: in EcZntB there are two Arg rings at positions 191 and 198

, but also such rings observed in all members - yes, they are much wider - but in theory binding 5 chloride ions in the wide pore vs binding one chloride ion in the narrow pore should yield similar effect.

Another issue - how can you be sure it is a chloride ion bound and not a water molecule? In Table 1 it can be seen that in two structures a few water molecules were modelled, so how was the distinction made? Regarding the 'barrier' role of chloride - the authors claim that the ion blocks the pathway - how easily the ion is knocked-out without any Mg²⁺ coming? Furthermore, to fully grasp the role of this, I believe the kinetics analysis is necessary, especially since the mutation of Arg yields no effect, so the only role I can imagine that indeed the chloride at this position might regulate the rate of transport - but additional experiments are necessary.

Furthermore - the mechanism of regulation is not clear - Mrs2 has the sites Mg-3 and Mg-4 which resemble the regulatory sites observed in TmCorA) and which are actually empty upon EDTA treatment (if I understood the passage at p.9 correctly). However the role of these sites absolutely unclear.

Minor issues:

Please avoid claims of high resolution - this is so badly abused in the field and my recommendation is to use the term for structures with the resolution better than 2 Å.

The manuscript will benefit from the correction by a native speaker / specialised service - in principle it is well written, but sometimes the choice of words is suboptimal, e.g. page 3: Mg²⁺ transmembrane transport is dependent on Mg²⁺ channels - in this case transport is not dependent on proteins but provided with or sustained;

p.3: dysfunction of Mg²⁺ balance -better Mg²⁺ imbalance;

page 4: data collection and data processes - should be data collection and data processing;

ultra-long helix - better to say extra long;

p.12 past findings - better to say previous studies; etc.

The title sounds a bit weird, instead of molecular understanding, I propose - molecular basis

the fact that EDTA has no impact on the structure was also observed in ZntB from E.coli;

page 6 - GMN motif is not just important, it is essential;

Fig.2 perhaps use 4I0U for T. maritima CorA as it has the loop modelled. Report Rmsd values!

Fig. 3 can be moved to supplementary.

Fig.4 indicate the sigma level of EM density

Fig.5b - please do not use red and green next to each other;

Table S1 - all but hMrs2-Mg structures have significantly larger clash score - can it be an indication that there is a slight deviation from the symmetry?

Fig S5 - remove the ligand from MjCorA cartoon.

Reviewer #2:

Remarks to the Author:

1. What are the noteworthy results?

Li et al report here the first structures of human Mrs2, a mitochondrial ion channel that transports Mg²⁺ to the mitochondrial matrix. Mrs2 is homologous to CorA, the prokaryotic Mg transporter. Previously, CorA crystal and cryo-EM structures have been solved in different conditions (high and low magnesium, different pHs), revealing the gating mechanism in the prokaryotic channel. However, no information was available for the eukaryotic channel. The four cryo-EM maps are of excellent quality and show pentameric structures with very similar features. They are also very similar to the prokaryotic channels, although in stark contrast with those, the different pH/Mg concentrations do not produce any conformational changes or loss of symmetry. One of the striking features reported in the paper is the apparent binding of Chloride ions.

2. Will the work be of significance to the field and related fields? How does it compare to the established literature? If the work is not original, please provide relevant references

The paper reports the first structure of an eukaryotic MRS2 channel, together with MD simulations; structures are very similar to the prokaryotic homologues. They also propose that Mg transport may depend on membrane potential: this was also suggested by a recent report by Day et al (<https://doi.org/10.1016/j.cell.2020.08.049>) where they discovered that lactate mobilizes Mg storage from the ER to the mitochondria, which is dependent on the membrane potential. The key findings with respect to the previous literature are 1) Cl binding in the center of the ion channel 2) Lack of conformational changes in response to different concentrations of Mg/EDTA.

The authors do not compare in depth the current structures with previously solved structures (although they show the figures). In page 6 they indicate "The hMrs2-Mg protomer is consistent with TmCorA, MjCorA and EcZntB in all general features, but differs greatly in structural details (Fig. 2c), which may indicate structural and functional conservatism between different members of the CorA family, but diversity in conformational regulation". This sentence is not very clear: from the text and figures presented it cannot be determined which are the main differences between the reported structures and the prokaryotic proteins, or any diversity in conformational regulation.

On the other hand, some sections are described in a lot of detail, for example the "subunit interaction"; are there any unique features of this channel in terms of interaction between subunits which provide information about the functional properties of the channel (and specially, any unique feature compared with the previously reported structures?)

3. Are there any flaws in the data analysis, interpretation and conclusions? Do these prohibit publication or require revision? Does the work support the conclusions and claims, or is additional evidence needed?

In the case of chloride binding, the authors should provide additional support for the identity of the anion (beside the MD simulations) which support the binding. Also, mutants on the R-ring residues do not seem to have any functional impact; a functional validation supporting the proposed dual role of Chloride both favoring permeation and blocking the channel is needed, as it is probably the most relevant aspect of the paper. How is R332 conserved in eukaryotic organisms?

On the other hand, the authors propose that no significant changes occur in the absence of Mg; do you see any Chloride binding in the absence of Mg? Is the ion conducting pathway identical with and without Mg?

4. Is the methodology sound? Does the work meet the expected standards in your field?

Cryo-EM maps are of excellent quality, authors have deposited both maps and pDBs, and report all the relevant cryo-EM data.

5. Is there enough detail provided in the methods for the work to be reproduced?

The authors should include:

- Which pCDNA plasmid was used for the cloning
- For transient transfection, what amount of DNA and PEI:DNA ratio used in transfection.
- What concentration of Flag peptide was used for elution?
- Specify the type of masks that were used for refinement.
- Specify how map sharpening was performed.

Also:

- What is the origin of the drop observed in all the FSC curves?
- Why DTT was only used in Buffer E?

6. Other minor comments:

- Please, in Figure 6, include all the relevant values in the legend, as well as statistical analysis.
- In page 11, first paragraph, correct: "Unexpectedly, the presence of Cl⁻ significantly reduce the energy"
- Use "Uniprot" instead of "Uniprot"
- For the Flag resin, "Genenscript" is written, are you referring to Genscript resin? Specify the used resin.
- Page 24: correct "the resolutions was evaluated"
- Page 27: correct "Mg²⁺ uptake assay buffer (AB) is consisted"

Responses to Reviewers

Reviewer #1 (Remarks to the Author):

In this manuscript Li et al., report the long-awaited structure of Mrs2 channel from CorA superfamily of proteins. This a great milestone for the field, and there are some interesting observations like the stability of the protein even after extensive EDTA treatment and the putative involvement of chloride ions. Nevertheless I have some concerns and suggestions to the authors, which can hopefully improve their manuscript to be considered for publication in Nature Communications.

Major issues:

1. The comparison of the obtained Mrs2 structures with the known structures should be extended also to CmaX protein doi: 10.1016/j.ijbiomac.2021.06.130 as it also has somewhat different geometry. It is advisable to provide at least rmsd values in the analysis.

Response: As suggested by the reviewer, a comparison of hMrs2 with PaZntB (the CmaX protein) has been included in the revised main text, revised Fig. 2, and revised Supplementary Fig. 5. The RMSD values for structural comparisons have been included in the revised manuscript.

2. The authors claim the unique role of arginines and chlorides - but this is not novel. This was already discussed back in 2009 by Tan et al., DOI: 10.1002/pro.215 where one of their main conclusions is that "the transportation of metal ions can be further enhanced by binding chloride ions that can significantly lower the electrostatic barrier across the pore".

Response: As pointed out by the reviewer, we rephrased the sentence in the Abstract section of the revised manuscript as follows "A special structural feature of Cl⁻-bound R-ring, which consists of five Arg332 residues, was found in the hMrs2 structure." and in the Discussion section of the revised manuscript as follows "the R-ring bound with Cl⁻ in hMrs2 forms a special structural element in certain members of the CorA family, suggesting that hMrs2 may have a different Mg²⁺ permeation mechanism than TmCorA.".

Chloride ions were previously found in the structures of EcZntB (Tan et al., Protein Science, 2009) and MjCorA (Eshaghi et al., Science, 2006). These chloride ions bind to the cytoplasmic domain not the transmembrane domain in the protein and may play an indirect role in regulating Mg²⁺ transport. In sharp contrast, the chloride ion in our hMrs2 structure was found in the transmembrane pore region. This chloride is located at the center of the five-fold

symmetry axis and is simultaneously coordinated by five arginine residues, which directly mediate the transport of Mg^{2+} . No similar structural unit has been found in CorA family members thus far.

3. Furthermore despite indeed the Arginine which the authors propose to bind chloride ions is not conserved in the sequence, most likely it has at least some structural conservation: in EcZntB there are two Arg rings at positions 191 and 198, but also such rings observed in all members - yes, they are much wider - but in theory binding 5 chloride ions in the wide pore vs binding one chloride ion in the narrow pore should yield similar effect.

Response: We thank the reviewer for the suggestion. We superimposed the structure of hMrs2 with that of TmCorA, MjCorA, EcZntB and PaZntB but did not find a structural unit similar to the arginine ring in the transmembrane pore region in the four structures.

As the reviewer mentioned, Arg191 and Arg198 in the EcZntB cytoplasmic domain form two rings inside the pore. The distance between the two guanidine groups on the opposite arginine residue exceeds 15.6 Å, which is too wide to enable chloride ion coordination by these groups. If five chloride ions were to bind simultaneously to the same arginine ring, then the negative charges of the chloride ions would repel each other, making this structure unlikely to be formed. In addition, we downloaded the map (EMD-3605) of EcZntB and did not find chloride ion density peaks near Arg191 or Arg198.

4. Another issue - how can you be sure it is a chloride ion bound and not a water molecule? In Table 1 it can be seen that in two structures a few water molecules were modelled, so how was the distinction made?

Response: We assigned a chloride ion to the densities portion of the R-ring center for several reasons: 1) Electrostatic balance. As shown in revised Fig. 3f, the distance between the guanidine groups of each arginine residue and the center of the density peak is approximately 3.2 Å. Five guanidine groups form a strong positively charged ring. An anion in this position counteracts the positive charges and thus balance the electrostatic field. In our protein purification system, chloride ions (150 mM) were the predominant negatively charged ions. 2)

Structural stability for MD simulations. As shown in revised Fig. 4d-e, during 500-ns MD simulations, a chloride ion is stably bound at the center of the R-ring, while water molecules move quickly along the entire pore. 3) Structural similarity to other channels. In the crystal structure of the hexameric Orai Ca^{2+} channel, residues lysine 163, lysine 159 and arginine 155 formed three layers of positively charged rings in the Ca^{2+} conduction pore (Hou et al., Science, 2012). At the center of the ring, an anion, not a water molecule, was confirmed by the X-ray anomalous Fourier map. In summary, our biased view is that the assignment of a chloride ion to the center of the arginine ring in our hMrs2 structure is a reasonable choice and most likely to reflect the true structure.

5. Regarding the 'barrier' role of chloride - the authors claim that the ion blocks the pathway - how easily the ion is knocked-out without any Mg^{2+} coming?

Response: As shown in the revised Fig. 4d, the chloride ion blocks the ion conduction pathway during 500-ns MD simulations, indicating that it is highly stable at the center of the R-ring. The probability of being knocked out is very low.

6. Furthermore, to fully grasp the role of this, I believe the kinetics analysis is necessary, especially since the mutation of Arg yields no effect, so the only role I can imagine that indeed the chloride at this position might regulate the rate of transport - but additional experiments are necessary.

Response: We thank the reviewer for the suggestions. As shown in the revised Fig. 5b, there may be other unidentified Mg^{2+} channels or transporters in the inner mitochondrial membrane, and this possibility limits the effectiveness of kinetics analyses based on purified mitochondria. We therefore reconstituted the purified hMrs2 wild-type protein into liposomes and evaluated Mg^{2+} conduction with or without valinomycin added. As shown in Fig. R1 below, we did not observe significant Mg^{2+} uptake in the liposome assay. We speculated that the liposomes may

not fully mimic the inner mitochondrial membrane, as some key factors may be absent, and thus cannot recapitulate Mg^{2+} passage through hMrs2 channels *in vitro*.

Fig. R1 fluorescent transport assays. Changes in fluorescent signal of the dye Magnesium Green during uptake of Mg^{2+} via hMrs2 channels reconstituted in liposomes under different conditions.

7. Furthermore - the mechanism of regulation is not clear - Mrs2 has the sites Mg-3 and Mg-4 which resemble the regulatory sites observed in TmCorA and which are actually empty upon EDTA treatment (if I understood the passage at p.9 correctly). However the role of these sites absolutely unclear.

Response: Yes, the Mg-3 and Mg-4 sites in the hMrs2-Mg structure are similar to the regulatory sites observed in MjCorA and are unoccupied after EDTA treatment. A structural comparison of hMrs2-Mg and hMrs2-highEDTA did not reveal any obvious conformational changes. In our hMrs2 system, Mg^{2+} bound to the intersubunit groove but may not be involved in regulation. We speculate that hMrs2 may play an additional role by functioning as a mitochondrial Mg^{2+} reservoir, since Mg^{2+} is important to mitochondrial function.

Minor issues:

8. Please avoid claims of high resolution - this is so badly abused in the field and my recommendation is to use the term for structures with the resolution better than 2 Å.

Response: As suggested by the reviewer, we delete the term “high resolution” throughout the paper.

9. The manuscript will benefit from the correction by a native speaker / specialised service - in principle it is well written, but sometimes the choice of words is suboptimal, e.g. page 3: Mg^{2+} transmembrane transport is dependent on Mg^{2+} channels - in this case transport is not dependent on proteins but provided with or sustained;

Response: As suggested by the reviewer, our manuscript has been edited by Professional Language Service. We modified the sentence in the revised manuscript as follows “Mg²⁺ transmembrane transport is sustained by Mg²⁺ channels and transporters”.

*10. p.3: dysfunction of Mg²⁺ balance -better Mg²⁺ imbalance;
page 4: data collection and data processes - should be data collection and data processing;
ultra-long helix - better to say extra long;
p.12 past findings - better to say previous studies; etc.*

Response: We thank the reviewer for nice suggestions. We modified them in the revised manuscript accordingly.

*11. The title sounds a bit weird, instead of molecular understanding, I propose - molecular basis
the fact that EDTA has no impact on the structure was also observed in ZntB from E.coli;
page 6 - GMN motif is not just important, it is essential;*

Response: We thank the reviewer for nice suggestions. We modified them in the revised manuscript accordingly. The sentence “Similar results have been previously reported for EcZntB, which maintained a symmetric pentameric state even after EDTA treatment” is included in Discussion section of the revised manuscript.

12. Fig.2 perhaps use 4I0U for T. maritima CorA as it has the loop modelled. Report Rmsd values!

Response: As suggested by the reviewer, the structure of TmCorA (RCSB 4I0U) was used in the revised Fig. 2 and Supplementary Fig. 5. Also, the C α rmsd values were included in the legend.

13. Fig. 3 can be moved to supplementary.

Response: As suggested by the reviewer, we moved the Fig. 3 to the Supplementary Fig. 6 in the revised manuscript.

14. Fig.4 indicate the sigma level of EM density

Response: As suggested by the reviewer, we included the sigma level of EM density into the revised figure legend as follows “The maps of Mg-1, Mg-2, Mg-3, Mg-4 and Cl-1 contoured at the 4.0 σ , 4.0 σ , 3.0 σ , 2.5 σ and 4.0 σ level, respectively”.

15. Fig.5b - please do not use red and green next to each other;

Response: As suggested by the reviewer, we changed the colors of the old Fig. 5b-c (red, green and blue) to the colors of the new Fig. 4b-c (red, blue and gray).

16. Table S1 - all but hMrs2-Mg structures have significantly larger clash score - can it be an indication that there is a slight deviation from the symmetry?

Response: We thank the reviewer for pointing out this error. We further improved the geometry of the four structures, and the clash score of hMrs2-Mg, hMrs2-lowEDTA, hMrs2-highEDTA and hMrs2-rest dropped to 5.46, 6.27, 4.04 and 2.5, respectively.

17. Fig S5 - remove the ligand from MjCorA cartoon.

Response: As suggested by the reviewer, we removed the ligand from MjCorA cartoon accordingly.

Reviewer #2 (Remarks to the Author):

1. What are the noteworthy results?

Li et al report here the first structures of human Mrs2, a mitochondrial ion channel that transports Mg²⁺ to the mitochondrial matrix. Mrs2 is homologous to CorA, the prokaryotic Mg transporter. Previously, CorA crystal and cryo-EM structures have been solved in different conditions (high and low magnesium, different pHs), revealing the gating mechanism in the prokaryotic channel. However, no information was available for the eukaryotic channel. The four cryo-EM maps are of excellent quality and show pentameric structures with very similar features. They are also very similar to the prokaryotic channels, although in stark contrast with those, the different pH/Mg concentrations do not produce any conformational changes or loss of symmetry. One of the striking features reported in the paper is the apparent binding of Chloride ions.

Response: We thank the review for positive assessment.

2. Will the work be of significance to the field and related fields? How does it compare to the established literature? If the work is not original, please provide relevant references.

The paper reports the first structure of an eukaryotic MRS2 channel, together with MD simulations; structures are very similar to the prokaryotic homologues. They also propose that Mg transport may depend on membrane potential: this was also suggested by a recent report by Day et al (<https://doi.org/10.1016/j.cell.2020.08.049>) where they discovered that lactate mobilizes Mg storage from the ER to the mitochondria, which is dependent on the membrane potential. The key findings with respect to the previous literature are 1) Cl binding in the center of the ion channel 2) Lack of conformational changes in response to different concentrations of Mg/EDTA.

Response: We thank the review for positive assessment.

1). The authors do not compare in depth the current structures with previously solved structures (although they show the figures). In page 6 they indicate “The hMrs2-Mg protomer is consistent with TmCorA, MjCorA and EcZntB in all general features, but differs greatly in structural details (Fig. 2c), which may indicate structural and functional conservatism between different members of the CorA family, but diversity in conformational regulation”. This sentence is not very clear: from the text and figures presented it cannot be determined which are the main differences between the reported structures and the prokaryotic proteins, or any diversity in conformational regulation.

Response: We thank the reviewer for nice suggestions. We modified the manuscript in three ways: a) we revised the sentence in the revised manuscript as follows “The general features of the hMrs2-Mg protomer are similar to those in TmCorA, MjCorA, EcZntB and PaZntB, which may indicate structural and functional conservation among different members of the CorA family, however two structural details of these proteins differ greatly (Fig. 2c and Supplementary Fig. 5)”; b) We included the main difference between the reported structures and the prokaryotic proteins in the revised manuscript as follows “The first difference involves the N-terminal α/β domain, in which the number of β -strands and α -helices differ, resulting in a large difference in the protomer–protomer interface of the N-terminus. The second involves the orientation of H9 relative to H8, which results in a different position of the selection filter GMN motif”; c) We included the RMSD value of the comparison between two structures in revised Fig. 2 legend.

2). On the other hand, some sections are described in a lot of detail, for example the “subunit interaction”; are there any unique features of this channel in terms of interaction between subunits which provide information about the functional properties of the channel (and specially, any unique feature compared with the previously reported structures?)

Response: We thank the reviewer for nice suggestions. We deleted the “Subunit interaction” section and moved the old Fig. 3 to the new supplementary Fig. 6 in the revised manuscript. The neck region in the supplementary Fig. 6F is very unique in the hMrs2 channel and describes the interactions involved in chloride bound R-ring, which may regulate Mg²⁺ conduction.

3. Are there any flaws in the data analysis, interpretation and conclusions? Do these prohibit publication or require revision? Does the work support the conclusions and claims, or is additional evidence needed?

In the case of chloride binding, the authors should provide additional support for the identity of the anion (beside the MD simulations) which support the binding.

Response: In the response to the review 1, we explained in detail the reason that we assigned the density of the R-ring to a chloride ion. Briefly, the R-ring forms an area with a very strong positive charge, and an anion is absolutely required to maintain the electrostatic balance. The chloride ion (150 mM) is the predominant negatively charged ion in our system. Moreover, in the Orai calcium channel, anion binding to the R-ring center has been similarly observed. Therefore, the assignment of the chloride ion to the R-ring center in our hMrs2 structure is a reasonable choice and most likely represents the true structure. Notably, the binding of chloride ions to the cytoplasmic domain of CorA family proteins has been previously reported (Eshaghi et al., Science, 2006; Tan et al., Protein Sci., 2009).

To our knowledge, crystal-clear evidence for chloride ion binding in the R-ring center require X-ray anomalous Fourier map. However, obtaining a diffractive hMrs2 crystal for obtaining an anomalous dataset is a very challenging task, which we will undertake in the future.

2) Also, mutants on the R-ring residues do not seem to have any functional impact; a functional validation supporting the proposed dual role of Chloride both favoring permeation and blocking the channel is needed, as it is probably the most relevant aspect of the paper.

Response: We thank the reviewer for the helpful suggestions. To our knowledge, MD simulation techniques are often used in channel studies to show dynamic ion conduction. As

shown in Fig. 4d-e in the revised manuscript, 500-ns MD simulations revealed that continuously redistributing water channels are blocked by a chloride ion (Fig. 4d), and they reform after manual removal of the chloride ion (Fig. 4e), suggesting that a chloride ion blocks the channel. Next, as shown in Fig. 4f and Fig. 5d in the revised manuscript, the presence of chloride favors Mg^{2+} permeation by counteracting the Mg^{2+} charge through the positively charged R-ring. Our biased view is that the MD simulation results presented in our manuscript support a dual role for chloride ions in Mg^{2+} conduction. In the future, we will continue to analyze the unique Mg^{2+} mechanism underlying hMrs2 channel conduction.

3) *How is R332 conserved in eukaryotic organisms?*

Response: As shown in Fig. 1F of the previous report (Uthayabalan et al., Life Sci Alliance, 6(4):e202201742, 2023), the residue R332 is conserved in eukaryotic organism.

4) *On the other hand, the authors propose that no significant changes occur in the absence of Mg; do you see any Chloride binding in the absence of Mg? Is the ion conducting pathway identical with and without Mg?*

Response: Yes. As shown in Supplementary Fig. 7, the Mg-1, Mg-2 and Cl-1 sites were clearly visible in the cryo-EM reconstruction of hMrs2-highEDTA. The ion conducting pathway is identical with and without Mg^{2+} . Similar results have been previously reported on EcZntB, where EDTA has no impact on the structure (Gati et al., Nat. Comm., 2017).

4. *Is the methodology sound? Does the work meet the expected standards in your field? Cryo-EM maps are of excellent quality, authors have deposited both maps and pDBs, and report all the relevant cryo-EM data.*

Response: We thank the review for positive assessment.

5. *Is there enough detail provided in the methods for the work to be reproduced?*

The authors should include:

1) - *Which pCDNA plasmid was used for the cloning*

Response: The pcDNA4/Myc-His A plasmid was used for the cloning in our experiment. The information has been included in “Plasmid construct” section of the revised manuscript.

2) - *For transient transfection, what amount of DNA and PEI:DNA ratio used in transfection.*

Response: We included the related information in the revised manuscript as follows “Transient transfection was performed using PEI 25k (Polysciences, Inc). For 1.0×10^9 cells, 1.0 mg of plasmid and 3 mg of PEI were preincubated in 90 mL of fresh medium for 20 minutes. Then, the DNA/PEI mixture was added to the cell cultures”.

3) - *What concentration of Flag peptide was used for elution?*

Response: The concentration of Flag peptide is 0.4 mg/mL for elution. We included this information in the revised manuscript as follows “The target protein was eluted with buffer C containing 0.4 mg mL^{-1} DYKDDDDK peptide”.

4) - *Specify the type of masks that were used for refinement.*

Response: We included the masks information in the revised manuscript as follows “The mask output at the non-uniform refinement step was used as the initial mask for the subsequent local refinement step. The local refinement procedure, integrating a dynamically generated mask in each refinement iteration, was executed in conjunction with non-uniform adaptive regulation”.

5) - *Specify how map sharpening was performed.*

Response: We included the map sharpening information in the revised manuscript as follows “The automatically sharpened map derived from the local refinement step was subsequently employed for subsequent model building and refinement”.

Also:

6) - *What is the origin of the drop observed in all the FSC curves?*

Response: The FSC curves exhibit a dip region at approximately 6 angstroms, which is attributed to the presence of detergent micelles. The unmasked FSC curve gives the impression of two overlapping curves, representing different resolutions. One curve corresponds to the lower resolution region encompassing the micelles, with an approximate resolution of 6 Å, while the other curve represents the higher resolution portion that is expected to reveal finer details in our protein structure.

7)- *Why DTT was only used in Buffer E?*

Response: We thank the reviewer for pointing out this error. Actually, 1 mM DTT was used in both buffer D and buffer E for all hMrs2 proteins. We revised the manuscript as follows: “Subsequently, the concentrated eluent was loaded onto a Superose 6 increase 10/300 GL column (Cytiva), which had been preequilibrated with buffer D (20 mM Tris-HCl pH 8.0, 150 mM NaCl, 1 mM DTT and 0.007% (w v⁻¹) GDN). For the hMrs2-Mg structure, 20 mM MgCl₂ was added to buffer D. For the hMrs2-lowEDTA structure, 1 mM EDTA was added to buffer D. For the hMrs2-highEDTA structure, buffer E (10 mM MES pH 6.8, 150 mM NaCl, 5 mM EDTA, 0.5 mM EGTA, 1 mM DTT and 0.007% (w v⁻¹) GDN) was used for gel filtration.”

6. *Other minor comments:*

1) *Please, in Figure 6, include all the relevant values in the legend, as well as statistical analysis.*

Response: As suggested by the reviewer, we included all the relevant values as well as statistical analysis in the legend of the revised Fig. 5 (old version Fig. 6).

2) - *In page 11, first paragraph, correct: “Unexpectedly, the presence of Cl⁻ significantly reduce the energy”*

Response: We corrected the sentence as “Unexpectedly, the Cl⁻ significantly reduced the free energy, thereby favoring Mg²⁺ permeation (Fig. 4f)” in the revised manuscript.

3) - *Use “Uniprot” instead of “Uniprot”*

Response: We used the term “UniProt ID: ” through the revised manuscript.

4) - *For the Flag resin, “Genenscript” is written, are you referring to Genscript resin? Specify the used resin.*

Response: We thank the reviewer for pointing out this error. The resin “Anti-DYKDDDDK Tag (L5) Affinity Gel (Biolegend)” was used in our protein purification system and the resin information was included the revised manuscript.

5) - *Page 24: correct “the resolutions was evaluated”*

Response: We corrected the sentence as “The resolution of the maps was evaluated using the “gold-standard” FSC=0.143 criterion” in the revised manuscript.

6) - *Page 27: correct “Mg²⁺ uptake assay buffer (AB) is consisted”*

Response: We corrected the sentence as “Mg²⁺ uptake assay buffer (AB) consisted of IB” in the revised manuscript.

Reviewers' Comments:

Reviewer #1:

Remarks to the Author:

I thank the authors for the revised manuscript which I believe became more logical and easier to follow.

I would still prefer the authors to tone down a bit their conclusions regarding the unique role of the chloride as there is no 100% evidence that the found sites are indeed occupied by chloride ions - I agree that placing the chloride there, is perhaps the most plausible explanation, but this is all based on assumptions and MD simulations. Otherwise the authors need to perform more biochemical experiments, e.g. in chloride-free conditions. The same goes for the authors reply in the rebuttal that the situation with 5 chloride ions in the pore is not realistic - one can argue that adding more Mg²⁺ ions and water perhaps can shield these repulsions - and this can be actually proved at least with MD; hence perhaps the authors should at least discuss that such chloride-assisted transport can not be completely ruled out for other members of CorA family at least for now

Reviewer #2:

Remarks to the Author:

Li et al report here the first structures of the human Mrs2 channel in different pH and EDTA concentrations; structures reveal chloride binding in the center of the ion conduction pathway. Further analysis by MD allows to propose a model for Mg permeation, where Cl would be helping Mg to cross the R-ring barrier; mutants on the R332 residue do not abolish transport, suggesting the Cl would have a regulatory role. One of the concerns was the identity of the ion; the authors compare now with the Orai channel and provide further explanations justifying the identity of the ion. While there are still some open questions (such as the role of the Mg3 and Mg4 binding sites), the authors have addressed most of the comments made by reviewers and the report of this structure is highly relevant for the field.

Minor comment:

1- sentence in Line 254 " we propose a scenario in which Mg²⁺ permeation is mediated via the hMrs2 channel". This sentence is not very clear, as you propose a model for permeation (the fact that Mg permeation is mediated via the Mrs2 channel was previously demonstrated).

Responses to Reviewers

Reviewer #1 (Remarks to the Author):

I thank the authors for the revised manuscript which I believe became more logical and easier to follow.

I would still prefer the authors to tone down a bit their conclusions regarding the unique role of the chloride as there is no 100% evidence that the found sites are indeed occupied by chloride ions - I agree that placing the chloride there, is perhaps the most plausible explanation, but this is all based on assumptions and MD simulations. Otherwise the authors need to perform more biochemical experiments, e.g. in chloride-free conditions. The same goes for the authors reply in the rebuttal that the situation with 5 chloride ions in the pore is not realistic - one can argue that adding more Mg²⁺ ions and water perhaps can shield these repulsions - and this can be actually proved at least with MD; hence perhaps the authors should at least discuss that such chloride-assisted transport can not be completely ruled out for other members of CorA family at least for now.

Response: As suggested by the reviewer, we have revised our manuscript to tone down our conclusion regarding the special role of the chloride as follows “A possible Cl-binding site, composed of five Arg332 residues (R-ring), was identified in hMrs2. Sequence alignment analysis indicated that the residue Arg332 was not conserved among members of the CorA family (Supplementary Fig. 5), implying that hMrs2 might employ a distinct Mg²⁺ permeation mechanism compared to TmCorA^{18,20,24,26}”.

Additionally, we have incorporated a new paragraph in the last part of the Discussion section as follows “It is worth noting that while the R-ring in the transmembrane pore region of the hMrs2 channel is not present in other members of the CorA family, wider arginine rings in the matrix domain have been observed in the structures of TmCorA, MjCorA, EcZntB, and PaZntB^{19,22,27,28}. Consequently, the possibility of chloride-assisted Mg²⁺ transport cannot be entirely disregarded for other members of the CorA family, at least for the current time being.”

Reviewer #2 (Remarks to the Author):

Li et al report here the first structures of the human Mrs2 channel in different pH and EDTA concentrations; structures reveal chloride binding in the center of the ion conduction pathway. Further analysis by MD allows to propose a model for Mg permeation, where Cl would be helping Mg to cross the R-ring barrier; mutants on the R332 residue do not abolish transport, suggesting the Cl would have a regulatory role. One of the concerns was the identity of the ion; the authors compare now with the Orai channel and provide further explanations justifying the identity of the ion. While there are still some open questions (such as the role of the Mg³ and

Mg4 binding sites), the authors have addressed most of the comments made by reviewers and the report of this structure is highly relevant for the field.

Minor comment:

1- sentence in Line 254 " we propose a scenario in which Mg²⁺ permeation is mediated via the hMrs2 channel". This sentence is not very clear, as you propose a model for permeation (the fact that Mg permeation is mediated via the Mrs2 channel was previously demonstrated).

Response: We thank the reviewer for nice suggestion. The sentence was changed accordingly in the revised manuscript as follows "we propose a model for Mg²⁺ permeation in the hMrs2 channel".